# Indirect Predictors of Visceral Adipose Tissue in Women with Polycystic Ovary Syndrome: A Comparison of Methods

**DOI:** 10.3390/nu13082494

**Published:** 2021-07-22

**Authors:** Małgorzata Kałużna, Magdalena Czlapka-Matyasik, Aleksandra Bykowska-Derda, Jerzy Moczko, Marek Ruchala, Katarzyna Ziemnicka

**Affiliations:** 1Department of Endocrinology, Metabolism and Internal Diseases, Poznan University of Medical Sciences, 60-355 Poznan, Poland; mruchala@ump.edu.pl (M.R.); kaziem@ump.edu.pl (K.Z.); 2Department of Human Nutrition and Dietetics, Poznan University of Life Sciences, 60-624 Poznan, Poland; magdalena.matyasik@up.poznan.pl (M.C.-M.); aleksandra.derda@up.poznan.pl (A.B.-D.); 3Department of Computer Science and Statistics, Poznan University of Medical Sciences, 60-806 Poznan, Poland; jmoczko@ump.edu.pl

**Keywords:** polycystic ovary syndrome (PCOS), visceral obesity, densitometry (DXA), waist-to-height ratio (WHtR), android obesity, anthropometry

## Abstract

Visceral adipose tissue (VAT) accumulation, is a part of a polycystic ovary syndrome (PCOS) phenotype. Dual-energy x-ray absorptiometry (DXA) provides a gold standard measurement of VAT. This study aimed to compare ten different indirect methods of VAT estimation in PCOS women. The study included 154 PCOS and 68 age- and BMI-matched control women. Subjects were divided into age groups: 18–30 y.o. and 30–40 y.o. Analysis included: body mass index (BMI), waist circumference (WC), waist-to-hip ratio (WHR), waist-to-height ratio (WHtR), waist/height 0.5 (WHT.5R), visceral adipose index (VAI), lipid accumulation product (LAP), and fat mass index (FMI). VAT accumulation, android-to-gynoid ratio (A/G), and total body fat (TBF) was measured by DXA. ROC analysis revealed that WHtR, WHT.5R, WC, BMI, and LAP demonstrated the highest predictive value in identifying VAT in the PCOS group. Lower cut-off values of BMI (23.43 kg/m^2^) and WHtR (0.45) were determined in the younger PCOS group and higher thresholds of WHtR (0.52) in the older PCOS group than commonly used. Measuring either: WHtR, WHT.5R, WC, BMI, or LAP, could help identify a subgroup of PCOS patients at high cardiometabolic risk. The current observations reinforce the importance of using special cut-offs to identify VAT, dependent on age and PCOS presence.

## 1. Introduction

Visceral adipose tissue (VAT) is a component of total body fat (TBF) responsible for high metabolic activity because it releases many bioactive hormones and molecules [1]. Women with polycystic ovary syndrome (PCOS) are at risk for visceral obesity (VO) [2,3]. PCOS is a complex, multifaceted disease linked with hyperandrogenism, hyperandrogenisation, oligoovulation, and polycystic ovarian morphology (PCOM) using ultrasound [4]. About half of the patients diagnosed with PCOS are obese, according to the classical World Health Organization (WHO)’s definition of body mass index (BMI) [5]. PCOS contributes to several metabolic and hormonal disorders which are favourable for increased VAT deposition. Research shows that abdominal fat is the most harmful for metabolic-related disorders [6]. An excessive accumulation of VAT is associated with hypertension, disturbed lipid profile, insulin resistance (IR), and hyperglycaemia [7]. Abdominal obesity is a significant contributor to metabolic syndrome (MS) development, cardiovascular diseases (CVD, obesity-associated malignancies, type 2 diabetes (T2D), and fatty liver [7,8]. Faced with these risks, the need to determine the best intermediate predictor of VO in PCOS is essential. Methods to evaluate VO have changed over the years. The most precise method is to utilise computed tomography (CT) or magnetic resonance imaging (MRI) to quantify VAT [9]. CT imaging and multi-slice volume MRI are the gold standard research implements for directly evaluating VO.

However, the mentioned methods are expensive, require sophisticated equipment, and are not appropriate for use in clinical practice [9]. Densitometry (DXA) VAT assessment remains the best indirect way to diagnose excess abdominal fat accumulation [9]. DXA-measured VAT volume strongly correlates with MRI-estimated VAT volume [10,11] and has stronger odds ratios for T2D and CVD vs. waist circumference (WC) [12]. The android-to-gynoid (A/G) ratio derived from DXA is valuable in assessing body fat distribution (BFD) and the prediction of VO [13]. DXA-derived total body fat (TBF) indirectly reflects both total and abdominal obesity [14]. The fat mass index (FMI) is a tool assessing body fat according to height, similarly to BMI for assessing body weight [15]. Despite the usefulness of DXA to measure VAT, the cost and size of the equipment may be an obstacle for clinical practice. Determining either the sole WC, a combination of circumferences waist-to-hip ratio (WHR), waist-to-height ratio (WHtR), or traditional BMI could be useful in assessing abdominal fat [5]. WC/Height^0.5^ (WHT.5R) has been proposed as an anthropometric index, considered as a good alternative to surrogate markers of visceral fat [16]. Recently used anthropometric and lipid-derived parameters, lipid accumulation product (LAP), and visceral adiposity index (VAI) are also affordable predictors of excess adiposity, insulin resistance (IR), and metabolic syndrome (MS) [17]. There is no general agreement on the most effective predictor of VO. Even though women with PCOS are at risk for VO, studies on this group and different VAT measurements are lacking and insufficient [11,18,19,20].

This study was designed to evaluate the most precise and accurate VAT predictors for women with PCOS by identifying the cut-off points for the various indirect indicators of VO. To date, these criteria have not been clearly characterised in PCOS patients.

## 2. Materials and Methods

### 2.1. DXA Measurements

The body composition and fat distribution were measured by the dual-energy x-ray absorptiometry (DXA) method with a Lunar Prodigy^TM^ (GE Healthcare^©^, Madison, WI, USA, 2013) densitometer. All DXA measurements were performed by a single, certified individual. Scans were performed in the morning in a fasted state. The quality control was performed according to the user manual on each day of the study visits. Subjects had to remove all metal parts of clothing and accessories. Body composition parameters: total body fat percentage (TBF), total lean mass [21], and fat distribution: VAT was analysed using the software enCORE^TM^ (version 17) and CoreScan^TM^ (GE Healthcare^©^, Madison, WI, USA). Reference standards for BFD and VAT were set according to the study by Ofenheimer et al. [22].

Additionally, more BFD and body composition parameters were calculated:

TBF—total body fat [%] $=\frac{fat\;mass\;\left(kg\right)}{total\;mass\;\left(kg\right)}*100$

FMI—fat mass index [kg/m^2^] $=\frac{fat\;mass\;\left(kg\right)}{height\;{\left(m\right)}^{2}}$ [22]

A/G—android-to-gynoid ratio [-] $=\frac{android\;fat\;mass\;\left(kg\right)}{gynoid\;fat\;mass\;\left(kg\right)}$

LMI—lean mass index [kg/m^2^] $=\frac{total\;lean\;mass\;\left(kg\right)}{height\;{\left(m\right)}^{2}}$ [22]

appendicular LMI [kg/m^2^] $=\frac{lean\;mass\;of\;four\;limbs\;\left(kg\right)}{height\;{\left(m\right)}^{2}}$ [22]

### 2.2. Anthropometric Measurements

Body height and weight were measured on the digital scale with light clothing and no shoes. The following formula was used to evaluate BMI:

BMI—body mass index [kg/m^2^] $=\frac{body\;mass\;\left(kg\right)}{height\;{\left(m\right)}^{2}}$

WC and hip circumference (HC) were measured according to WHO guidelines and resulted in calculations of the following parameters [23]:

WHR—waist-to-hip ratio [-] $=\frac{waist\;circumference\;\left(m\right)}{hip\;circumference\;\left(m\right)}$

WHtR—waist-to-height ratio [-] $=\frac{waist\;circumference\;\left(m\right)}{height\;\left(m\right)}$

WHt.5R— $=\frac{waist\;circumference\;\left(m\right)}{height\;{\left(m\right)}^{0.5}}$ [16]

All anthropmetric measurements were performed by a single, experienced dietitian.

### 2.3. Study Population

A total of 222 women of reproductive age (18–40 y.o.) with (*n* = 154) and without (*n* = 68) PCOS were enrolled to the study between June 2016 and September 2019 and included in this analysis. The flowchart depicting patient selection process is shown in Figure 1. All patients underwent full interview, laboratory test, and physical examination, including gynecological consultation and transvaginal ultrasound. PCOS was diagnosed according to the 2018 international guidelines [24] and Rotterdam criteria, defined by the presence of at least two out of the following features: 1. oligoovulation/anovulation; 2. hyperandrogenism (elevated androgen concentration: total testosterone (T) > 2.67 nmol/L, and/or free androgen index (FAI): > 5.5) or hyperandrogenisation (acne, hirsutism); 2. ultrasound imaging of at least 20 follicles in each ovary measuring 2–9 mm in diameter, and/or ovarian volume more than 10 cm^3^ [4,24]. Control subjects (CON) were amenorrhoeic with no history of menstrual dysfunction or chronic disease and were BMI-, age-, and sex-matched with the PCOS population. The matching process was based on the method of M:1, in which patients in the PCOS group were matched with one of the counterparts in the CON group according to age and BMI. The patients were matched according to the closest distance with the replacement of the matched control subject [25]. Matching process resulted in exclusion of 51 healthy women (Figure 1) with an average age: 33.2 ± 6.6 y.o., and BMI: 21.5 ± 3.6.

All subjects with decompensated thyroid dysfunction, severe acute or chronic renal or liver diseases, Cushing’s disease or using oral contraceptives, hormonal replacement therapy, ovulation-inducing agents or anti-androgens over the past three months were excluded from the study.

Informed and written consent was obtained from all participants. The clinical examination protocol complied with the Declaration of Helsinki for Human and Animal Rights and its later amendments and received ethical approval from the Board of Bioethics of University of Medical Science (552/16; 986/17).

### 2.4. Laboratory Tests

Blood samples were collected from all participants after an overnight fast. Insulin, follicle-stimulating hormone (FSH), luteinising hormone (LH), dehydroepiandrosterone sulphate (DHEAS), oestradiol (E2), total testosterone (T), sex-hormone-binding globulin (SHBG), and anti-Müllerian hormone (AMH) were analysed using a Cobas 6000 (Roche Diagnostics, GmbH, Mannheim, Germany). Kits available from the manufacturer were used. Appendix A represents the lower detection limits of hormones assessed by electrochemiluminescence immunoassay (ECLIA). The free testosterone index (FTI) was determined by the formula: (FTI) = 100 × (total testosterone/SHBG). Total cholesterol (TC-C), high-density lipoprotein cholesterol (HDL-C), and triglycerides (TG-C) were evaluated by the enzymatic colorimetric method. The Friedewald formula was used to estimate Low-density lipoprotein cholesterol (LDL-C). Serum glucose was measured by the hexokinase method (Roche Diagnostics) with a coefficient of variation (CV) of 3%. The homeostasis model assessment for insulin resistance (HOMA-IR) and the formula: HOMA-IR = (fasting plasma glucose (mg/dL) × fasting plasma insulin (mU/L))/405 were used. HOMA-IR > 2.5 was used as the threshold to determine IR.

### 2.5. Statistical Analysis

The statistical software used for analysis was Statistica v.13.1 (*StatSoft* Polska sp. z o.o., Kraków, Poland). The Shapiro–Wilk test was used to examine the distribution of continuous variables. Descriptive statistics for quantitative variables were presented as the median and interquartile range (IQR). Differences in measured parameters between PCOS and CON groups were calculated using an independent sample *t*-test when the variances were homogeneous (tested with a Levine test) or an independent sample *t*-test with separate variance estimates. A Mann–Whitney U test was applied for significantly skewed data. Pearson’s linear correlation coefficients were used to explore the linear association between visceral obesity indices and other parameters. Two-tailed *p* values of <0.05 were determined as statistically significant.

Tentative cut-points were derived of VAT for the best VO predictor for classifying individuals as centrally obese for females under 30 years old 235.6 g and for females 30 and over up to 40 years old 340.3 g [22]. Patients were divided into four groups according to age; 1. PCOS < 30 y.o. (*n* = 122), 2. PCOS 30–40 y.o. (*n* = 32), 3. CON < 30 y.o. (*n* = 49), 4. CON 30–40 y.o. (*n* = 19).

Receiver operating characteristic (ROC) curve analysis was used to establish the clinical usefulness and optimal cut-off values for obesity indices in predicting VAT. The optimal prediction threshold is defined as the cut-off point with the maximum Youden index (sensitivity + specificity − 1).

## 3. Results

Descriptive data for the anthropometric, biochemical, and hormonal variables in PCOS and CON groups are shown in Table 1. The clinical characteristics of the age groups of PCOS and CON subjects are shown in Appendix A. PCOS patients had lower concentrations of TSH and higher levels of LH, T, FTI, and AMH. Parameters of simple and central obesity were not significantly different among PCOS and CON women (Table 1). No significant differences in VAT mass were found in different weight categories (normal-weight, overweight and obese) between PCOS patients and controls, respectively.

Table 2 shows Pearson’s correlations between VAT, weight, age, and all selected VO predictors. The strongest (*r* > 0.7) correlations were observed between VAT and WHtR, WHT.5R, WC, LAP, BMI, weight, and FMI. All ten selected VO markers were significantly correlated with each other (*p* < 0.001). The weakest correlations were observed in the A/G ratio and TBF compared to other anthropometric parameters (Table 2). After dividing the PCOS patients into obese (BMI ≥ 30 kg/m^2^) and non-obese, all anthropometric indicators correlated with VAT in the non-obese group (*p* < 0.05), while in the obese group, VAT correlated positively only with BMI, WC, WHR, WHtR, LAP, WHT5R (*r* = 0.422, *r* = 0.816, *r* = 0.417, *r* = 0.821, *r* = 0.537, *r* = 0.825, respectively).

The prevalence of obesity in PCOS and CON based on classical cut-offs of selected obesity predictors are presented in Table 3. There were no significant differences between the obesity frequency in PCOS and CON subjects using any obesity indicator (*p* > 0.05), except for the A/G ratio. The lowest incidence of obesity was estimated based on BMI (PCOS vs. CON: 20.3 vs. 14.3%). The highest VO prevalence was estimated from WHR (PCOS vs. CON 64.0 vs. 64.4%), followed by the DXA-derived VAT (PCOS vs. CON < 30 y.o.: 53.3 vs. 36.7%; PCOS vs. CON 30–40 y.o.: 56.2 vs. 77.8%) and TBF (PCOS vs. CON: 50.3 vs. 47.0%) (Table 3).

Non-parametric ROC analysis showed the predictive ability of the VO indices in the age groups of PCOS and CON subjects. WHtR had the greatest area under curve (AUC) in both age groups of PCOS women (PCOS < 30: AUC 0.954 (95% CI 0.921–0.986, *p* < 0.001); PCOS 30–40 y.o.: AUC 0.973 (95% CI, *p* < 0.001)) (Table 4). The statistical significance of the AUCs differences in PCOS 18–30 y.o. showed that the AUC of WC (0.953, LAP (0.947), WHT.5R (0.946) and BMI (0.917) were not significantly different than the AUC of WHtR (*p* > 0.05) (Appendix A). Similarly, WHT.5R, WC, BMI, FMI, LAP and VAI had similar to WHtR predictive value in VO prognosis when considering AUC differences in PCOS30–40 y.o. (*p* > 0.05). The optimal cut-off values (sensitivity, specificity) of VO predictors are shown in Table 4.

In both age groups of CON, ROC analysis showed that the AUC of BMI (18–30 y.o.: 0.890; 30–40 y.o.: 1.00) were the highest in the younger group followed by WC (0.875), WHT.5R (0.855), WHtR (0.839), in the older group followed by WC (0.985), FMI (0.929), and WHT.5R (0.924). (Appendix A). In younger CON, only WHR from all other VO predictors had significantly lower predictive power than BMI (*p* = 0.003). In the older group of CONs, there was no significant difference between AUC of BMI and FMI (*p* > 0.05) (Appendix A). The sensitivity (specificity) of BMI and FMI in this CON group was 100% (100%) and 93% (100%), respectively (Table 4).

## 4. Discussion

VO is an important health issue linked to several severe metabolic complications and serious long-term implications [8]. VO seems to be associated with a chronic inflammation, stress, and unfavourable metabolic profile in both PCOS and general population [26,27,28]. There is a continuous debate which comes first: obesity or PCOS [29]. Increased systematic inflammation can be the consequence of visceral obesity and PCOS itself [27]. Central and simple obesity may or may not occur together [30]. Variation of VAT accumulation depends on gender, age, ethnicity, smoking status, physical activity level, alcohol intake, and genetic predisposition [7,31]. The prevalence of VO is typically higher than the spread of simple obesity [30]. For comparison, the prevalence of obesity in the current sample, as estimated by using the BMI, was about 2.5 times lower than using VAT (Table 3). Quantitative evaluation of VO is essential for assessing the possible cardiometabolic risk, especially in PCOS women [9]. Calculating VAT from DXA or other specialised equipment such as MRI is the most precise, but it is expensive, time-consuming, expensive, and not widely available [9,10,11]. It is crucial to identify an indirect, simple tool that can quickly and accurately delineate VAT and monitor changes over time.

BFD differs between PCOS and healthy women; however, data on VAT excess in PCOS is inconsistent [18,20]. Some studies have found increased estimated VAT in PCOS patients, whether obese or non-obese than age- and BMI-matched controls [18,20]. There was no significant difference in the amount of VAT between consecutive, unselected PCOS and CON subjects in the current study. No significant differences in VAT mass were found in any weight category between PCOS and CON women. On the contrary, in the Carmina et al. study, overweight and normal weight, but not obese PCOS women had a higher quantity of central abdominal fat in DXA vs. BMI-matched controls [32]. All obese patients, with or without PCOS, had increased VAT in that study [32]. Small increases in visceral obesity can be simply reflected in changes in anthropometric measures [32]. The simple correlation analysis shows that in obese patients, BMI, WC, WHR, WHtR, LAP, WHT5R appears to reflect better VAT than weight, VAI, A/G ratio, FMI, or TBF in the current study. The above observations coincide with the results of the ROC analysis in the entire PCOS group, which selected BMI, WC, WHtR, LAP, WHT5R as the most effective VAT predictors.

The incidence and severity of obesity depended on the race, ethnicity, and type of patient selection. The incidence of obesity and severe obesity and mean BMI were similar in unselected PCOS subjects and the general population in the Ezeh et al. study [33].

As far as it is known, this is the only study in which ROC curves were used to assess the ten obesity indicators for VO screening in reproductive-aged women. The results suggest that WHtR, WHT.5R, WC, BMI, and LAP appear to have similar, outstanding (AUC ≥ 0.9) prognostic value in VO prediction in both PCOS age groups in the Hosmer and Lemeshow classification system for AUC [34]. In the CON group aged over 30 y.o., BMI was the best anthropometric predictor of VO. The observed differences in the strength of VAT predictors between PCOS and a healthy population could result from hormonal differences between the groups. Hyperandrogenism has been linked to severe metabolic consequences, including IR, VO and MS [35].

Another outcome of the current study was determining the cut-offs with the best trade-off between true-positive and false-positive rates which varied with PCOS status and age. Lower cut-off values for BMI (23.43 kg/m^2^) and WHtR (0.45) were noted in the younger PCOS group than officially used. Higher cut-off values for WHtR (0.52) in the older PCOS group than widely approved (0.5) were also identified. Cut-off points for predicting VO was also lower in the current study than previously recommended [16]. In a study by Swanson et al., VO was predicted to be 0.59 for both sexes [16]. The same cut-off value for WC as the WHO cut-off in the younger PCOS group (80.0) and higher values in the older PCOS group (85.0) was observed [5]. In conclusion, the risk of VO seems to be increased in young women with PCOS and the anthropometric parameters within the norms for the general population. Age, population, and disease-specific cut-offs should be used for VO prediction in women of reproductive age. One other study supports the current results and shows that the cut-off value of 80.0 for WC in women with PCOS is a good predictor of cardiovascular risk [36].

All analysed anthropometric obesity parameters (BMI, WC, WHR, WHtR, and WHT.5R) are clinically accessible, require little equipment, and are quick, repetitive, and inexpensive [37]. The current study emphasises the role of four of them (BMI, WC, WHtR, and WHT.5R) in the effective prediction of VAT mass in PCOS. WHR was the only anthropometric indicator with unacceptable (AUC < 0.7) prognostic VO power in PCOS aged 18–40 y.o. In line with the current study, Borruel et al. suggested that WHR is a poor indicator of VO (even worse than WC) [38]. Other studies suggest that WHtR and combinations of WHtR with BMI are better in predicting cardiometabolic risk than WHR or WC with BMI [39].

Although BMI, the official measurement to diagnose obesity, has well-known limitations, it is still valuable in population screening [5,40]. The discovery of the usefulness of BMI in the whole sample (equal to or even better than other predictors) in the current study was unexpected. BMI reflects TBF without regard to fat distribution although it may fail to precisely indicate body fatness in specific groups, e.g., highly trained people or patients with cachexia [41]. However, the usefulness of BMI was proven in the PCOS population [42]. Ortega et al. claimed that BMI is a more valuable predictor of CVD mortality than total adiposity markers [40]. Their results obtained using an analysis of 60,000 adults suggested that an excess of body weight is related to a worse CVD prognosis than an excess of TBF [40]. In women, BMI appears to be a more accurate body fat predictor than WHR [43]. However, in the present study only around 1 in 5 PCOS women were obese according to the BMI criterion versus around half of each sample according to the VAT thresholds. The use of only BMI for the assessment may be misleading and should be further studied in the context of PCOS.

Evidence also suggests that WC is a valuable predictor of VO, even better than WHR, especially in young women [38,43]. The current results support this thesis. Ethnicity-specific values for WC are available to distinguish adults at increased cardiometabolic risk and the range of WC shifts significantly for women (80–96) [44]. Unfortunately, WC measuring is prone to location errors. The WHO recommends the midpoint between the last palpable rib and the iliac crest and the National Institutes of Health recommends the level of the umbilicus. Additionally, the WHO recommends focusing on WC for the prediction of cardiometabolic disease only with BMI below 35 kg/m^2^ [45]. In a study by Raimi et al., WC and BMI had comparable accuracy in the prediction of body fat [43]. WC and BMI seem to be the best surrogate markers of VO in young adults in both sexes [38]. Janssen et al. proved that BMI and WC independently contribute to the prognosis of VO [42]. As mentioned previously, WC thresholds for VO screening had the same WC cut-off value as WHO reference in the younger PCOS group (80.0) and higher in the older PCOS group (85.0) [5]. Moreover, studies using representative populations are required to establish age-, ethnicity-WC threshold values, and the best anatomic location of measurement in PCOS patients [44].

Several studies provide strong clues of WHtR usefulness in VO and MS prediction [46]. The results of the Swainson et al. study suggest that WHtR is the best predictor of TBF and VAT, independently of sex [16]. Cut-offs for predicting whole body obesity were 0.53 in men and 0.54 in women and for VO it was 0.59 in both sexes [16]. In a recent cohort based on data from the Health Survey of England, WHtR seemed to predict better cardiometabolic disease than BMI and WC [47]. The current study supports these findings by showing that WHtR is a good predictor of VAT.

In the current study, WHT.5R had comparable power to WHtR for VAT prediction. In a study by Swanson et al., WHT.5R was found to be a good predictor of VAT in both sexes [16]. However, neither the current study nor the Swanson et al. study confirmed the advantage of additional calculations of this parameter over the less complicated WHtR [16].

TBF appears to be less predictive of VO than simple anthropometric parameters according to the current study and previous observations [9]. FMI seems to be far more prognostic of CVD mortality than TBF [40] since FMI had higher accuracy in VAT prediction than TBF and its power was higher in women aged 30–40 than in the younger group.

VAT and LAP are parameters that may help account for the estimation of risk of VO, IR and MS [19]. LAP seems to be more accurate in VAT prediction than VAI both in the current study and recent observations [19]. LAP also seems to better predict MS than VAI [17].

The current observations reinforce the suitability of measuring simple anthropometric parameters (especially BMI, WC, WHtR) to predict and monitor the risk of VO development. The disadvantage of the current study was the use of DXA-derived VAT, which is only an indirect measure. Even though some studies show the disadvantages of using DXA to measure VAT (no distinction between types of adipose tissues, underestimating VAT at low levels, overestimating VAT at high levels) [9,10], DXA-derived VAT remains a very accurate and precise method for visceral adipose tissue assessment [11]. Ideally validated by MRI or CT, future studies are needed in the longitudinal evaluation of surrogate VAT predictors.

## 5. Conclusions

In summary, WHtR, WHT.5R, WC, BMI, and LAP can be used with reasonable success to detect VO in women with PCOS. Changes in anthropometric parameters can simply represent small increases in visceral adiposity. Anthropometric parameters such as WHtR, WC, and BMI can diagnose VO in PCOS and should be widely used in basic patient assessment. The involvement of anthropometric variables in providing comprehensive instant results will enhance VO diagnosis and help to identify patients who need dietary and medical intervention. However, more studies on the accuracy and cut-offs of surrogate anthropometric measures of VO in different ethnic and age groups of PCOS women are still needed.

## Figures and Tables

**Figure 1 nutrients-13-02494-f001:**
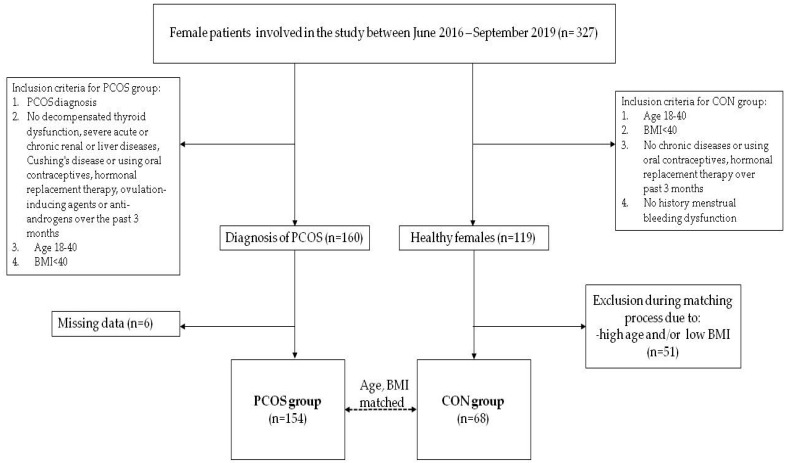
Study participant selection process.

**Table 1 nutrients-13-02494-t001:** General comparison between study (PCOS) and control (CON) samples (median ± interquartile range (IQR)).

Variable	PCOS	CON	*p* Value
	*n* = 154	*n* = 68	
Age (y.o.)	25.0 ± 7.0	24.1 ± 7.9	NS
Weight (kg)	67.50 ± 21.00	65.00 ± 19.00	NS
BMI (kg/m^2^)	23.88 ± 8.02	23.05 ± 6.58	NS
WC (cm)	78.00 ± 20.00	74.00 ± 15.00	NS
WHR (-)	0.88 ± 0.10	0.89 ± 0.08	NS
WHtR (-)	0.47 ± 0.12	0.45 ± 0.09	NS
WHT.5R (-)	0.61 ± 0.16	0.58 ± 0.11	NS
A/G ratio (-)	0.35 ± 0.21	0.35 ± 0.19	NS
VAT mass (g)	285.35 ± 602.25	214.26 ± 538.03	NS
TBF (%)	35.41 ± 0.12	34.22 ± 0.12	NS
FMI (kg/m^2^)	8.27 ± 5.00	8.22 ± 4.01	NS
LAP (-)	15.92 ± 28.48	13.56 ± 14.87	NS
VAI (-)	0.84 ± 0.95	0.93 ± 0.76	NS
SBP (mmHg)	120.00 ± 19.00	122.00 ± 14.00	NS
DBP (mmHg)	75.50 ± 11.00	76.00 ± 14.00	NS
Glucose (mg/dL)	89.00 ± 9.00	87.00 ± 10.00	NS
Insulin (µU/mL)	9.06 ± 6.91	9.45 ± 4.97	NS
HOMA-IR	1.99 ± 1.70	1.98 ± 1.06	NS
TC (mg/dL)	179.00 ± 35.00	166.00 ± 40.00	NS
TG (mg/dL)	71.00 ± 49.00	75.00 ± 51.00	NS
HDL-C (mg/dL)	63.00 ± 21.00	64.00 ± 26.00	NS
LDL-C (mg/dL)	95.45 ± 41.10	83.90 ± 36.90	NS
TSH (μU/mL)	2.02 ± 1.29	2.39 ± 1.76	**
FSH (mIU/mL)	5.95 ± 2.35	5.60 ± 3.50	NS
LH (mIU/mL)	8.65 ± 8.40	6.80 ± 5.60	*
E2 (pg/mL)	42.50 ± 40.00	68.00 ± 58.00	NS
T (nmol/L)	1.60 ± 0.90	1.20 ± 0.90	***
DHEAS (µg/dL)	305.00 ± 167.00	266.00 ± 165.00	*
SHBG (nmol/L)	54.30 ± 37.40	55.00 ± 43.60	NS
FTI (%)	3.15 ± 3.08	2.46 ± 2.25	*
AMH (pmol/L)	53.31 ± 36.52	23.35 ± 16.87	***

*** for *p* < 0.001, ** for *p* < 0.01, * for *p* < 0.05, NS for not statistically significant. AMH—anti-Müllerian hormone; BMI—body mass index; CON—control subjects; DBP—diastolic blood pressure; DHEAS—dehydroepiandrosterone sulfate; E2—estradiol; FPG—fasting plasma glucose; FSH—follicle-stimulating hormone; FTI—free testosterone index; LH—luteinising hormone; HDL-C—high-density lipoprotein cholesterol; HOMA-IR—homeostasis model assessment for insulin resistance index; LAP—lipid accumulation product; LDL-C—low-density lipoprotein cholesterol; PCOS—polycystic ovary syndrome patients; SBP—systolic blood pressure; SHBG—sex hormone-binding globulin; T—total testosterone; TC—total cholesterol; TG—triglycerides; VAI—visceral adiposity index; WC—waist circumference; WHtR—waist-to-height ratio.

**Table 2 nutrients-13-02494-t002:** Correlation analysis between visceral obesity predictors in PCOS women (*n* = 154).

Variable	BMI	WC	WHR	WHtR	WHT.5R	A/G Ratio	VAT	TBF	FMI,	LAP
	*r*, *p* Value	*r*, *p* Value	*r*, *p* Value	*r*, *p* Value	*r*, *p* Value	*r*, *p* Value	*r*, *p* Value	*r*, *p* Value	*r*, *p* Value	*r*, *p* Value
age	0.147,	0.191,	0.151,	0.20,	0.192,	0.142,	0.209,	0.103,	0.127,	0.225,
	NS	*	NS	*	*	NS	**	NS	NS	**
weight	0.930,	0.903,	0.423,	0.839,	0.866,	0.377,	0.745,	0.449,	0.766,	0.825,
	***	***	***	***	***	***	***	***	***	***
BMI	-	0.875,	0.405,	0.882,	0.901,	0.368,	0.784,	0.465,	0.829,	0.806,
		***	***	***	***	***	***	***	***	***
WC	0.875,	-	0.667,	0.966,	0.980,	0.434,	0.834,	0.449,	0.735,	0.922,
	***		***	***	***	***	***	***	***	***
WHR	0.405,	0.667,	-	0.664,	0.650,	0.250,	0.479,	0.117,	0.275,	0.632,
	***	***		***	***	0.002	***	0.16	***	***
WHtR	0.882,	0.966,	0.664,	-	0.977,	0.441,	0.859,	0.467,	0.745,	0.907,
	***	***	***		***	***	***	***	***	***
WHT.5R	0.901,	0.980,	0.650,	0.977,	-	0.423,	0.838,	0.448,	0.763,	0.903,
	***	***	***	***		***	***	***	***	***
A/G ratio	0.368,	0.434,	0.250,	0.441,	0.424,	-	0.472,	0.763,	0.670,	0.388,
	***	***	**	***	***		***	***	***	***
VAT mass	0.784,	0.834,	0.479,	0.859,	0.838,	0.472,	-	0.537,	0.726,	0.810,
	***	***	***	***	***	***		***	***	***
TBF	0.465,	0.449,	0.117,	0.467,	0.448,	0.763,	0.537,	-	0.857,	0.406,
	***	***	0.16	***	***	***	***		***	***
FMI	0.829,	0.735,	0.275,	0.745,	0.763,	0.670, ***	0.726,	0.857,	-	0.668,
	***	***	***	***	***		***	***		***
LAP	0.806,	0.922,	0.632,	0.907,	0.903,	0.388,	0.810,	0.406,	0.668,	-
	***	***	***	***	***	***	***	***	***	
VAI	0.525,	0.642,	0.492,	0.645,	0.620,	0.293,	0.646,	0.298,	0.446,	0.846,
	***	***	***	***	***	***	***	***	***	***

A/G ratio—android-to-gynoid ratio; BMI—body mass index; CON—control subjects; FMI—fat mass index, LAP—lipid accumulation product; PCOS—polycystic ovary syndrome patients; TBF—total body fat percentage, VAI—visceral adiposity index; VAT—visceral adipose tissue mass; WC—waist circumference; WHR—waist-to-hip ratio; WHtR—waist-to-height ratio, *** for *p* < 0.001, ** for *p* < 0.01, * for *p* < 0.05, NS for not statistically significant.

**Table 3 nutrients-13-02494-t003:** Presence of simple obesity and visceral obesity in PCOS and CON women based on classical cut-offs of selected visceral adiposity indicators.

Indicator of Visceral Obesity	PCOS	CON	*p* Value
	% (*n* = 154)	% (*n* = 68)	
BMI > 30 kg/m^2^	20.3%	14.3%	NS
WC ≥ 80 cm	44.2%	32.8%	NS
WHR > 0.85	64.0%	64.4%	NS
WHtR ≥ 0.5	36.7%	31.7%	NS
FMI ≥ 9.7 kg/m^2^ [22]	37.7%	32.8%	NS
TBF > 35%	50.3%	47.0%	NS
A/G ratio > 0.3 [22]	63.6%	59.7%	0.03
VAT mass *	18–30 y.o.—53.3%	18–30 y.o.—36.7%	NS
	30–40 y.o.—56.2%	30–40 y.o.—77.8%	NS

* 18–30 y.o. > 235.6 g, 30–40 y.o. > 340.3 g [22].

**Table 4 nutrients-13-02494-t004:** Selected parameters for predicting visceral obesity and the corresponding AUCs, optimal cut-off values, their sensitivity and specificity, and Youden index.

Variable	AUCs	Optimal Cut-Off Values	Sensitivity	Specificity	Youden Index
PCOS ^18–30^					
BMI	0.917	23.43	0.86	0.89	0.75
WC	0.953	80	0.82	0.96	0.78
WHR	0.783	0.89	0.66	0.79	0.49
WHtR	0.954	0.45	0.9	0.86	0.76
WHT.5R	0.946	0.59	0.87	0.86	0.76
A/G ratio	0.737	0.4	0.6	0.84	0.45
TBF	0.764	0.36	0.71	0.77	0.49
FMI	0.87	8.06	0.79	0.82	0.62
LAP	0.947	16.44	0.87	0.96	0.85
VAI	0.844	0.94	0.68	0.95	0.63
PCOS ^30–40^					
BMI	0.952	27.34	0.83	1	0.83
WC	0.958	85	0.88	1	0.88
WHR	0.681	0.97	0.44	1	0.44
WHtR	0.973	0.52	0.94	1	0.94
WHT.5R	0.969	0.66	0.94	1	0.94
A/G ratio	0.861	0.43	0.77	0.86	0.69
TBF	0.777	0.39	0.71	0.86	0.56
FMI	0.937	7.92	0.94	0.79	0.73
LAP	0.942	29.49	0.75	1	0.75
VAI	0.862	1.47	0.75	0.93	0.68
CON ^18–30^					
BMI	0.89	23.05	0.82	0.86	0.69
WC	0.875	79	0.77	0.96	0.73
WHR	0.639	0.87	0.71	0.61	0.31
WHtR	0.839	0.48	0.65	0.93	0.63
WHT.5R	0.855	0.61	0.71	0.96	0.67
A/G ratio	0.669	0.36	0.65	0.71	0.36
TBF	0.718	0.38	0.65	0.74	0.39
FMI	0.832	7.7	0.88	0.72	0.6
LAP	0.821	15.73	0.69	0.93	0.61
VAI	0.72	1.23	0.56	0.93	0.49
CON ^30–40^					
BMI	1	23.46	1	1	1
WC	0.985	80	0.91	1	0.91
WHR	0.727	0.88	1	0.67	0.67
WHtR	0.819	0.51	0.58	1	0.58
WHT.5R	0.924	0.64	0.71	1	0.73
A/G ratio	0.714	0.36	0.79	0.75	0.54
TBF	0.732	0.39	0.5	1	0.5
FMI	0.929	7.32	0.93	1	0.93
LAP	0.879	11.19	1	0.67	0.67
VAI	0.848	1.55	0.64	1	0.64

A/G ratio—android-to-gynoid ratio; BMI—body mass index; CON—control subjects; FMI—fat mass index, LAP—lipid accumulation product; PCOS—polycystic ovary syndrome patients; TBF—total body fat percentage, VAI—visceral adiposity index; VAT—visceral adipose tissue mass; WC—waist circumference; WHR—waist-to-hip ratio; WHtR—waist-to-height ratio.

## Data Availability

The data supporting the conclusions of this article are included within the article and its additional files. The other datasets used and/or analysed during the current study are available from the corresponding author upon reasonable request.

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
