# Peer review of "Indirect Predictors of Visceral Adipose Tissue in Women with Polycystic Ovary Syndrome: A Comparison of Methods"

_nutrients, 2021, doi:10.3390/nu13082494_

Round 1

Reviewer 1 Report

This is an interesting paper confirming that simple anthropometric measures may be useful for assessing visceral obesity.

However, most studied patients were not obese and in these patients (as reported in JCEM 2007: 92:2500-2505) mild increases of visceral obesity may be easily reflected in chenages of anthropometric measures. It may be much more diffciult to get the same correlations in obese patients, mainly in patints with moderate or severe obesity.

The authors should reanalyze the data, separating obese patients from normal weight or overweight patients and check whether the correlations are still present.

Minor: references are not complete and more studies on visceral fat, lean mass and dexa should be included.

Author Response

            Dear Reviewer,

Thank you very much for all your comments and suggestions. We truly believe the visceral obesity is an important topic that is often not fully recognized. It is true that the correlations between patients with and without simple obesity often differ. As we mentioned in Discussion section, data on VAT excess in PCOS is inconsistent [1,2]. Some studies have found increased estimated VAT in PCOS patients, whether obese or non-obese than age- and BMI-matched controls [1,2]. There was no difference in the amount of VAT between consecutive, unselected PCOS and CON subjects in the current study. We reanalyzed the data and separated obese, overweight and normal weight patients.    No significant differences in VAT mass were found in different weight categories (normal-weight, overweight and obese) between PCOS patients and  controls, respectively. On the contrary, in Carmina et al. study overweight and normoweight, but not obese PCOS women had higher quantity of central abdominal fat in DXA vs. BMI-matched controls [3]. All obese patients, with or without PCOS, had increased VAT in that study [3]. The incidence and severity of obesity depended on the race, ethnicity and type of patient selection. The incidence of obesity and severe obesity and mean BMI were similar in unselected PCOS subjects and the general population [4].

After dividing the PCOS patients into obese (BMI≥30 kg/m2) and non-obese, all anthropometric indicators correlated with VAT in the non-obese group (p<0.05), and in the obese group, VAT correlated (p<0.05) positively only with BMI, WC, WHR, WHtR, LAP, WHT5R (r=0.422, r=0.816, r=0.417, r=0.821, r=0.537, r=0.825, respectively). Small increases of visceral obesity may be simply reflected in changes of anthropometric measures. However, in obese patients, BMI, WC, WHR, WHtR, LAP, WHT5R better reflect VAT than weight, VAI, A/G ratio, FMI or TBF. The above observations coincide with the results of the ROC analysis in the entire PCOS group, which selected BMI, WC, WHtR, LAP, WHT5R as the most effective VAT predictors.

Due to the small size of the group of patients with PCOS and BMI 30 (31 patients in total), and the fact that only 1 of 31 patients with PCOS and simple obesity did not meet the diagnosis of visceral obesity, the ROC analysis of selected VAT predictors cannot be technically performed in subgroup of obese PCOS patients. From a practical point of view, it seems that the prediction of VAT and visceral obesity is more important in the groups of patients not initially suspected of visceral obesity, i.e. with BMI within the norm. We are planning further studies on subpopulations of PCOS patients and both low and high BMI, where we will definitely use the above advice and the cited work. We cited the  study by Carmina et al. (JCEM 2007: 92:2500-2505) in our manuscript. The references were checked and corrected. We added a few citations of the interesting studies. Unfortunately, there is no to much reliable studies on visceral obesity measured by densitometry in PCOS. Thank you once again for all the valuable and accurate suggestions.

  1. Jena, D.; Choudhury, A.K.; Mangaraj, S.; Singh, M.; Mohanty, B.K.; Baliarsinha, A.K. Study of Visceral and Subcutaneous Abdominal Fat Thickness and Its Correlation with Cardiometabolic Risk Factors and Hormonal Parameters in Polycystic Ovary Syndrome. Indian J Endocrinol Metab 2018, 22, 321-327, doi:10.4103/ijem.IJEM_646_17.
  2. Satyaraddi, A.; Cherian, K.E.; Kapoor, N.; Kunjummen, A.T.; Kamath, M.S.; Thomas, N.; Paul, T.V. Body Composition, Metabolic Characteristics, and Insulin Resistance in Obese and Nonobese Women with Polycystic Ovary Syndrome. J Hum Reprod Sci 2019, 12, 78-84, doi:10.4103/jhrs.JHRS_2_19.
  3. Carmina, E.; Bucchieri, S.; Esposito, A.; Del Puente, A.; Mansueto, P.; Orio, F.; Di Fede, G.; Rini, G. Abdominal fat quantity and distribution in women with polycystic ovary syndrome and extent of its relation to insulin resistance. J Clin Endocrinol Metab 2007, 92, 2500-2505, doi:10.1210/jc.2006-2725.
  4. Ezeh, U.; Yildiz, B.O.; Azziz, R. Referral bias in defining the phenotype and prevalence of obesity in polycystic ovary syndrome. J Clin Endocrinol Metab 2013, 98, E1088-1096, doi:10.1210/jc.2013-1295.
  5. Teede, H.J.; Misso, M.L.; Costello, M.F.; Dokras, A.; Laven, J.; Moran, L.; Piltonen, T.; Norman, R.J.; International, P.N. Recommendations from the international evidence-based guideline for the assessment and management of polycystic ovary syndrome. Hum Reprod 2018, 33, 1602-1618, doi:10.1093/humrep/dey256.

Reviewer 2 Report

The authors have performed a significant study that has a lot to add to the current knowledge of the management in PCOS patients. In fact, clinical medicine needs more predictive variables for assessing visceral adipose tissue in these patients. Minor concerns of mine:

  1. The Methods section need amelioration. The authors need to explicitly report the inclusion criteria one by one in the manuscript. Who did the measurements? Was there only one researcher, who measured the patients, or more? If more, please provide the k aggreement.
  2. There are missing references also. For example, PCOS is an acknowledged subclinical inflammatory entity. How does the metabolic stress in PCOS link to the visceral obesity? this detail should be reported. 'Chronic stress and body composition disorders: implications for health and disease by Stefanaki et al. 2018 Mar;17(1):33-43.'.
  3. The authors should also report on the significance of visceral adiposity and its implications in their Discussion section. Perna et al. Osteosarcopenic Visceral Obesity and Osteosarcopenic Subcutaneous Obesity, Two New Phenotypes of Sarcopenia: Prevalence, Metabolic Profile, and Risk Factors. J Aging Res. 2018; 2018: 6147426.

Author Response

Dear Reviewer, thank you for all the valuable and accurate suggestions. Visceral adiposity is still unevaluated issue in diagnostics and treatment of polycystic ovary syndrome (PCOS). As we confirmed, that simple anthropometric measures may be useful for assessing visceral obesity in patients with PCOS .

  1. The Method section was improved. We added study subject selection process flowchart as well as explained exact procedure of matching. The inclusion criteria were revised. PCOS was diagnosed according to the latest 2018 international guidelines [1] and Rotterdam criteria, defined in the text [1,2]. Control subjects (CON) were amenorrhoeic with no history of menstrual dysfunction or chronic disease and were BMI-, age- and sex-matched with the PCOS population. All patients underwent full interview, laboratory test and physical examination, including gynecological consultation and transvaginal ultrasound. The exlusion criteria were also clearly stated.  A single researcher, experienced MD, (MK) qualified patients as PCOS and control subjects according to the criteria given above. One qualified dietitian performed all simple anthropometric measurements and body composition assessment by densitometry (A.B-D.).
  2. The references were checked and corrected. Chronic stress and subclinical inflammation are linked with metabolic disorders, especially insulin resistance and visceral obesity in both PCOS and health subjects.  We added the citation of manuscript by Sefanaki et. al. [3]. There is still a debate which obesity or PCOS comes first. Increased inflammation can be the consequence of visceral obesity in PCOS, rather than PCOS itself [4].
  3. We added the data on significance of visceral obesity and its implications in Discussion section. The citation of study  by Perna et. al. was added [5]. Although there is a single observation on polycystic ovary syndrome as a risk factor for sarcopenic obesity by McBrearity et al., we did not confirmed the presence of sacropenic obesity in our group of patients with PCOS based on appendicular lean mass index/m2 and reference values for our densitometry scaner (unpublished data) (GE Prodigy, Ofenheimer et al. (2020)). Bone and mass mass correlate positively with higher androgens in PCOS. We are planning next studies on body composition and nutritional status of women with PCOS.

Thank you once more time for your commitment and time.  

  1. Teede, H.J.; Misso, M.L.; Costello, M.F.; Dokras, A.; Laven, J.; Moran, L.; Piltonen, T.; Norman, R.J.; International, P.N. Recommendations from the international evidence-based guideline for the assessment and management of polycystic ovary syndrome. Hum Reprod 2018, 33, 1602-1618, doi:10.1093/humrep/dey256.
  2. Rotterdam, E.A.-S.P.C.W.G. Revised 2003 consensus on diagnostic criteria and long-term health risks related to polycystic ovary syndrome. Fertil Steril 2004, 81, 19-25.
  3. Stefanaki, C.; Pervanidou, P.; Boschiero, D.; Chrousos, G.P. Chronic stress and body composition disorders: implications for health and disease. Hormones (Athens) 2018, 17, 33-43, doi:10.1007/s42000-018-0023-7.
  4. Kaluzna, M.; Czlapka-Matyasik, M.; Wachowiak-Ochmanska, K.; Moczko, J.; Kaczmarek, J.; Janicki, A.; Piatek, K.; Ruchala, M.; Ziemnicka, K. Effect of Central Obesity and Hyperandrogenism on Selected Inflammatory Markers in Patients with PCOS: A WHtR-Matched Case-Control Study. J Clin Med 2020, 9, doi:10.3390/jcm9093024.
  5. Perna, S.; Spadaccini, D.; Nichetti, M.; Avanzato, I.; Faliva, M.A.; Rondanelli, M. Osteosarcopenic Visceral Obesity and Osteosarcopenic Subcutaneous Obesity, Two New Phenotypes of Sarcopenia: Prevalence, Metabolic Profile, and Risk Factors. J Aging Res 2018, 2018, 6147426, doi:10.1155/2018/6147426.

Round 2

Reviewer 1 Report

The authors have improved the manuscript analyzing the differences between obese and non obese subjects.